# Enhancing Efficiency in Inverted Quantum Dot Light-Emitting Diodes through Arginine-Modified ZnO Nanoparticle Electron Injection Layer

**DOI:** 10.3390/nano14030266

**Published:** 2024-01-26

**Authors:** Young-Bin Chae, Su-Young Kim, Hyuk-Doo Choi, Dae-Gyu Moon, Kyoung-Ho Lee, Chang-Kyo Kim

**Affiliations:** Department of Electronic Materials, Devices and Equipment Engineering, Soonchunhyang University, Asan 31538, Republic of Korea; w200r23@sch.ac.kr (Y.-B.C.); swim8549@sch.ac.kr (S.-Y.K.); hyukdoo.choi@sch.ac.kr (H.-D.C.); dgmoon@sch.ac.kr (D.-G.M.)

**Keywords:** arginine, PEI, interlayer, ZnO nanoparticles, work function inverted quantum dot light-emitting diode

## Abstract

Many quantum dot light-emitting diodes (QLEDs) utilize ZnO nanoparticles (NPs) as an electron injection layer (EIL). However, the use of the ZnO NP EIL material often results in a charge imbalance within the quantum dot (QD) emitting layer (EML) and exciton quenching at the interface of the QD EML and ZnO NP EIL. To overcome these challenges, we introduced an arginine (Arg) interlayer (IL) onto the ZnO NP EIL. The Arg IL elevated the work function of ZnO NPs, thereby suppressing electron injection into the QD, leading to an improved charge balance within the QDs. Additionally, the inherent insulating nature of the Arg IL prevented direct contact between QDs and ZnO NPs, reducing exciton quenching and consequently improving device efficiency. An inverted QLED (IQLED) utilizing a 20 nm-thick Arg IL on the ZnO NP EIL exhibited a 2.22-fold increase in current efficiency and a 2.28-fold increase in external quantum efficiency (EQE) compared to an IQLED without an IL. Likewise, the IQLED with a 20 nm-thick Arg IL on the ZnO NP EIL demonstrated a 1.34-fold improvement in current efficiency and a 1.36-fold increase in EQE compared to the IQLED with a 5 nm-thick polyethylenimine IL on ZnO NPs.

## 1. Introduction

Quantum dots (QDs) have emerged as a distinctive and advanced category of emissive materials, exhibiting exceptional material properties, such as high color saturation, emission wavelengths, a narrow spectral line width, compatibility with solution-based processing, and robust stability in various environmental conditions [1,2,3,4,5,6]. These inherent characteristics have significantly facilitated the development of QD-based devices, particularly in quantum dot light-emitting diodes (QLEDs), which exhibit unique and captivating optoelectronic characteristics [7,8,9,10,11]. Ongoing research in this field has greatly improved QLED performance. Remarkably, state-of-the-art QLEDs have achieved an external quantum efficiency (EQE) of 20% [2,10]. Consequently, QLED performance is rapidly approaching parity with organic light-emitting diodes (OLEDs) [12]. Recently, there has been extensive research on perovskite light-emitting didoes for next-generation display [13,14].

Inverted QLEDs (IQLEDs), first reported in 2012 [15], have attracted significant attention due to their advantage of connecting bottom cathodes to n-type thin film transistor backplanes, facilitating the integration of active matrix QLEDs [16,17,18]. Furthermore, in the IQLED, the direct deposition of solution-processed ZnO nanoparticles (NPs) on the cathode ensures that it does not affect other layers produced through various solution processes [19]. However, a challenge arises in the injection and transport of holes from anode to the QD emission layer (EML), which encounters a higher energy barrier compared to the injection and transport from electron injection layers (EILs) to the QD EML. This imbalance in charge (both holes and electrons) within the QD EML has been observed [20,21,22]. The imbalance results in higher turn-on voltage and reduced device efficiency. Additionally, commonly used electron transport layers (ETLs) like ZnO NPs exhibit higher electron mobility than the hole mobility of hole transport layers (HTLs). Consequently, rectifying the excess electron injection into the QD EML stands as a crucial step in enhancing device efficiency. Zhang et al. reported that the IQLED with tandem structure exhibited an impressive current efficiency of EQE over 23% [23]. In 2016, Kim et al. introduced an IQLED with a tandem structure that allows for reducing damage to the upper functional layer during the solution process [23]. In 2017, Zhang et al. attained a current efficiency surpassing 100 cd A^−1^ and an impressive external EQE exceeding 23% through the utilization of Poly (3,4-ethylenedioxythiophene):poly(styrene sulfonate)/ZnMgO as the interconnecting layer [24]. Wu et al. introduced a semiconductor-metal-dielectric stack as an interconnect layer, successfully mitigating solvent damage during multiple solution treatments. Consequently, the tandem device achieved a groundbreaking EQE of 40% for red, 49% for yellow, and 24% for blue [23]. 

The ZnO NPs exhibit n-type properties, attributed to interstitial Zn atoms and the oxygen vacancies [25,26]. These characteristics make them widely used as the ETL in QLEDs, offering advantages such as high mobility and compatible interface energy levels [27,28]. However, ZnO NPs often contain a significant concentration of oxygen vacancies, leading to nonradiative recombination with QDs [29,30,31]. Direct contact between the QD EML and ZnO NP ETL causes spontaneous electron transfer at the interface, resulting in exciton quenching [29,30]. The surfaces of ZnO NPs synthesized in air are known for their abundant hydroxyl group (-OH), acting as active sites. These sites induce exciton quenching, significantly impacting overall performance [31]. Therefore, addressing the interface between the QD EML and ZnO NP ETL is crucial to enhancing device performance. Dai et al. reported a red QLED with a low turn-on voltage of 1.7 V, a high peak EQE of 20.5%, and a long lifetime of over 100,000 h at an initial brightness of 100 cd/m^2^ [2]. The enhanced performance is ascribed to the poly(methyl methacrylate) (PMMA) insulating layer between the QD EML and ZnO NP ETL, effectively optimizing charge-injection balance or preventing the exciton quenching.

Considerable attention has focused on the properties of interlayers (ILs) like polyethylenimine (PEI) and its derivative, polyethylenimine-ethoxylated (PEIE), primarily due to their exceptional ability to facilitate electron injection. Zhou et al. reported the efficacy of surface ins derived from polymers containing simple aliphatic amine groups, such as PEI and PEIE. These modifiers effectively reduce the work function of various conductive materials, including metals, transparent conductive metal oxides, and conducting polymers [32]. The intensity of an interfacial dipole in the modifiers is attributed to the concentration of protonated amines [N^+^]. These amines induce electrostatic dipoles, and a sufficient quantity of carbon atoms [C], which interrupt the self-assembly of protonated amines in a polymer modifier [33]. These electrostatic dipoles significantly lower the work function of ZnO NPs, consequently reducing the energy barrier between the EIL and EML. Park et al. demonstrated that OLEDs employing two different types of PEIs, either linear or branched, effectively decrease the work function of ZnO NPs [34]. Additionally, Kim et al. observed that both PEIE and PEI play crucial roles in reducing the electron injection barrier between ZnO NPs and the EML. This reduction occurs by decreasing the work function of the underlying ZnO NPs, thereby effectively enhancing electron injection into the EML. Their findings revealed that the work function of ZnO NPs, when coated with PEI, is lower compared to when coated with PEIE. This distinction is attributed to the higher [N^+^]/[C] ratio of PEI [33]. However, despite an increase in dipole due to higher thickness, the current density gradually decreased. This decline occurred due to the growing inherent insulating property of the modifier, offsetting the rise in the work function. Ding et al. reported that PEI has a dual effect on electron injection, acting as a barrier due to its inherent insulating properties while simultaneously facilitating electron injection by reducing the work function of ZnO NPs/PEI [17]. In recent years, optoelectronic devices incorporating arginine (Arg) as an IL between the QD EML and ZnO NP ETL have been reported [35,36]. Most small molecules possess the property of forming self-assemble monolayers (SAMs) on a metal oxide surface through covalent bonding. SAMs are known to influence the work function through polar functional groups capable of playing a role in Ils [37]. Acidic groups commonly used as SAMs, such as phosphoric acid, carboxyl acid, and amino groups, necessitate the use of toxic solvents during the preparation and treatment processes. In contrast, amino acids offer the inherent benefits of environment-friendly characteristics dissolved in water solution, cost-effectiveness, and widespread availability [36]. Importantly, they feature hydrophilic carboxyl groups, making them soluble in water solutions, while simultaneously possessing polar amino groups that facilitate the formation of robust hydrogen bonds [35]. Consequently, amino acids can be uniformly deposited on metal oxides. We chose arginine as the interlayer in QLEDs because, among amino acids, it has the highest proportion of amine groups, creating electrostatic dipoles on the ZnO surface. Additionally, the amine group is advantageous for being environmentally friendly and cost-effective. The introduction of polar amino groups in the Arg can induce an interface dipole moment, consequently reducing the work function of ZnO NPs. Li et al. incorporated L-Arg into inverted organic solar cells as the ETL, resulting in improved device performance by modifying the work function and enhancing interface conductivity. Notably, the device’s lifetime significantly extended with ZnO NPs/L-Arg used as a double ETL, compared to using pristine ZnO NPs [30]. Li et al. also demonstrated that self-assembled monolayer-modified ZnO NP EILs using Arg notably enhanced electron injection efficiency in inverted OLEDs. ZnO NPs/Arg EILs exhibited an exceptionally low work function of 2.35 eV, lower than the ZnO NP EIL modified by PEI with a work function of 2.77 eV. As a result, green phosphorescent IOLEDs demonstrated a low turn-on voltage of 3.5 V, a maximum current efficiency of 59.1 cd/A, and a maximum EQE of 16.8% [36]. This modification in the work function holds a promising potential to enhance the efficiency of OLEDs, QLEDs, and organic solar cells.

In this paper, we successfully introduced a water-soluble Arg IL for IQLED applications. Our findings suggest that the performance improvement in IQLEDs can be attributed to several factors. The introduction of the Arg IL leads to the formation of a dipole moment in the Arg molecules, increasing the work function of the ZnO NP EIL, and enhancing its inherent insulating properties. We investigated the modified work function and interaction of the ZnO NP EIL by the Arg IL using ultraviolet photoelectron spectroscopy (UPS) and X-ray photoelectron spectroscopy (XPS). This modification aids in improving the charge balance within the QD EML. The significant enhancement in performance is attributed to the decreased electron injection, resulting from increased energy barriers and inherent insulating properties introduced by the Arg IL. Furthermore, the introduction of the Arg IL suppressed QD charging and reduced the presence of hydroxyl groups that typically influence exciton quenching. Additionally, the thick Arg IL deposited on ZnO NP contributes to a more uniform surface morphology by effectively filling the valleys at the surface of ZnO NP film. Our study demonstrates a charge-injection-balanced IQLED achieved by incorporating a relatively thick Arg IL between the ZnO NP EIL and QD EML. The resulting optimized current efficiency and EQE are 36.67 cd/A and 8.92%, respectively. These values represent 2.2-fold and 2.3-fold increases compared to an IQLED without the Arg IL. 

## 2. Materials and Methods

### 2.1. Synthesis of Materials

ZnO NPs in a colloidal suspension were synthesized using a sol–gel method, as described in our previous study [38]. To synthesize ZnO NPs, 0.3292 g of zinc acetate dihydrate (Zn(CH_3_COO)∙2H_2_O) powder (Sigma-Aldrich, St. Louis, MO, USA), serving as the precursor material, was dissolved in 45 mL of dimethyl sulfoxide (DMSO, Sigma-Aldrich). Additionally, 0.421 g of tetramethylammonium hydroxide (TMAH, Sigma-Aldrich) was dissolved in 15 mL of EtOH (Daejung Chemicals and Metals, Siheung, Republic of Korea). The TMAH and Zn(CH_3_COO)∙2H_2_O solutions were stirred at room temperature for 3 min to ensure complete dissolution. Ethyl acetate (Kanto Chemical Co., Inc., Tokyo, Japan) was added to the solution to precipitate the ZnO NPs at a volume ratio of 3:1. After 24 h, a white powder had formed and precipitated, and the ZnO NPs were obtained through centrifuging the solution. 

### 2.2. Device Fabrication

QLEDs with an inverted stacked structure, featuring a bottom cathode and a top anode, were fabricated. The basic device structure included an indium tin oxide (ITO) cathode (150 nm)/ZnO NP EIL (50 nm)/Arg (5 nm, 10 nm, 20 nm, 30 nm) or PEI (5 nm)/QD EML (10 nm)/4,4′,4-Tris (carbazol-9-yl) triphenylamine (TCTA) HTL (40 nm)/WO_x_ hole injection layer (HIL) (10 nm)/Ag anode (100 nm). To investigate the interfacial effect of Arg ILs, Arg with varying thicknesses were inserted as an IL between the QD EML and ZnO NP EIL. For device fabrication, a 50 nm-thick ZnO NP EIL was deposited onto an ITO cathode via spin-coating of a solution containing ZnO NPs. To prepare the Arg solution, 10 mg of L-Arg (Sigma-Aldrich) was dissolved in 10 mL of de-ionized water. To prepare the PEI solution, 3 mg of PEI (Sigma-Aldrich) was dissolved in 10 mL of ethyl alcohol. The Arg solution was heated at 60 °C for 2 h before being applied to the ITO/ZnO NP EIL. After spin-coating the Arg and PEI solutions onto the ZnO NP EIL, the ITO/ZnO NP EIL/(Arg or PEI) substrate was annealed at 100 °C for 10 min. To deposit the QD EML, CdSe/ZnS QDs (Zeus) were dissolved in heptane at a concentration of 5 mg/mL. This solution was then spin-coated onto the ITO/ZnO NP EIL/(Arg or PEI) substrate. The ITO/ZnO NP EIL/(Arg or PEI)/QD EML was annealed in air at 60 °C for 10 min. The QD EML had a thickness of 10 nm. TCTA (Luminescence Technology Corp., New Taipei City, Taiwan) HTL was deposited using a vacuum evaporation method with a working pressure of 4.0 × 10^−7^ Torr and the deposition rate was slowly increased from 0.4 Å to 1.0 Å. Subsequently, the WO_x_ HIL and Ag anode were deposited sequentially using a vacuum thermal evaporation method (working pressure 4.0 × 10^−7^ Torr and a deposition rate of 0.1 nm/s) when depositing the WO_x_ (iTASCO, Seoul, Republic of Korea) HIL and Ag (iTASCO, Seoul, Republic of Korea) anode. The patterns of these three layers were defined using shadow metal masks.

### 2.3. Characterizations

X-ray diffraction (XRD; D/Max-2200pc; Rigaku, Tokyo, Japan) was employed to investigate the formation of the ZnO NPs, utilizing Cu-Kα radiation applied to the centrifuged solution. Field-emission transmission electron microscopy (FE-TEM) (Tecnai F30 S-Twin; JEOL Ltd., Tokyo, Japan) was used to determine the actual particle size of the ZnO NPs. Atomic force microscopy (AFM) (XE7; Park Systems, Suwon, Gyeonggi, Republic of Korea) was employed to characterize the surface morphology and roughness of ZnO NPs, ZnO NPs/Arg, and ZnO NPs/PEI thin films. UPS, featuring X-ray photoelectron spectroscopy (PHI Quantera-II; Ulvac-PHI, Chigasaki, Kanagawa, Japan) with a He (I) (21.22 eV) gas discharge lamp, was used to characterize the work function and VBM of ZnO NPs, ZnO NPs/Arg, and ZnO NPs/PEI. XPS measurements were also conducted using the same system with UPS to characterize the Zn 2p, N 1s, and O 1s levels of ZnO NPs, ZnO NPs/Arg, and ZnO NPs/PEI. Time-resolved photoluminescence (TRPL) (FL-QM, Horiba, Kyoto, Japan) characteristics and photoluminescence (PL) (FP6500; JASCO, Tokyo, Japan) spectra of QD and QDs composed of ZnO NPs, ZnO NPs/PEI (5 nm) IL, and ZnO NPs/Arg (20 nm) IL were analyzed to study the exciton quenching properties.

For current density–voltage–luminance (J–V–L) measurement, a computer-controlled source meter (2400; Keithley Instruments, Cleveland, OH, USA) and luminance meter (LS100; Konica Minolta, Tokyo, Japan) were utilized. Electroluminescence (EL) spectra were recorded using a spectroradiometer (CS1000; Konica Minolta, Maunouchi, Chiyoda, Japan).

## 3. Results and Discussion

To assess the impact of the Arg on the device performance, IQLEDs were fabricated using different EILs of ZnO NPs (50 nm)/Arg (0 nm, 5 nm, 10 nm, 20 nm, and 30 nm). Additionally, for comparative analysis, IQLEDs with EILs composed of ZnO NPs (50 nm)/PEI (5 nm) were fabricated. Figure 1 presents the structures of IQLEDs with Arg and PEI ILs between the QD EML and ITO/ZnO NP EIL and chemical structures of Arg and PEI. Control devices were established using ITO/ZnO NPs and ITO/ZnO NPs/PEI EILs. Figure 2a shows the characteristic 2θ XRD patterns of ZnO NPs, revealing the crystal structure. The crystal structure exhibited the characteristic hexagonal wurtzite pattern typical of ZnO (Joint Committee on Powder Diffraction Standards [JCPDS] Card 1-1136). XRD analysis of ZnO NPs revealed reflections in the (100), (002), (101), (102), (110), (103), and (112) planes, all corresponding to ZnO. Slight shifts in the diffraction peaks and intensities of ZnO NPs were observed, indicating minor lattice distortions and alterations in the interatomic spacing. Figure 2b illustrates a representative FE-TEM image depicting ZnO NPs, with an average diameter measured at 4.29 nm.

To enable a more effective comparison with the intended development of the Arg IL, reference devices were created: specifically, IQLEDs incorporating PEI ILs with thicknesses of 3 nm, 5 nm, 7 nm, and 10 nm. Appendix A illustrates the EL characteristics of these IQLEDs. In Appendix A, the current densities measured at 9 V for IQLEDs with 3 nm-thick, 5 nm-thick, 7 nm-thick, and 10 nm-thick PEI ILs on ZnO NPs were 743.7 mA/cm^2^, 809.9 mA/cm^2^, 192.9 mA/cm^2^, and 213.1 mA/cm^2^, respectively. Among these, the IQLED with a 5 nm-thick PEI IL exhibited the highest current density. Appendix A shows the maximum luminance values for IQLEDs with 3 nm-thick, 5 nm-thick, 7 nm-thick, and 10 nm-thick PEI ILs, calculated to be 249,617 cd/m^2^, 270,995 cd/m^2^, 39,926 cd/m^2^, and 12,290 cd/m^2^, respectively. IQLEDs with 7 nm-thick and 10 nm-thick PEI ILs exhibited a significant decrease in luminance, which was attributed to their low current densities. Appendix A illustrates the maximum current efficiencies for IQLEDs with PEI ILs with thicknesses of 3 nm, 5 nm, 7 nm, and 10 nm, measured at 27.35 cd/A, 27.87 cd/A, 16.61 cd/A, and 4.55 cd/A, respectively. Similarly, Appendix A shows the maximum EQEs for these IQLEDs with 3 nm-thick, 5 nm-thick, 7 nm-thick, and 10 nm-thick Arg ILs, calculated at 6.57%, 6.70%, 4.10%, and 0.84%, respectively. The IQLED with a 5 nm-thick PEI IL was selected as the reference device for comparison with IQLEDs with various thicknesses of Arg ILs, because it exhibited the highest efficiency among IQLEDs with 3 nm-thick, 5 nm-thick, 7 nm-thick, and 10 nm-thick PEI ILs. The key parameters of IQLEDs with ZnO NPs/PEI (3 nm), ZnO NPs/PEI (5 nm), ZnO NPs/PEI (7 nm), and ZnO NPs/PEI (10 nm) EILs are summarized in Appendix A.

To investigate the influence of Arg and PEI ILs on the work function variation of the ZnO NP EIL, UPS measurements were conducted. The electronic energy level configuration of thin films, specifically ITO/ZnO NPs, ITO/ZnO NPs/PEI (5 nm), ITO/ZnO NPs/Arg (5 nm), ITO/ZnO NPs/Arg (10 nm), ITO/ZnO NPs/Arg (20 nm), and ITO/ZnO NPs/Arg (30 nm), was examined using UPS. Figure 3a,b illustrate UPS spectra corresponding to the secondary electron cutoff and valence band maximum regions. The estimated work functions below the vacuum level for the thin films are as follows: ITO/ZnO NPs (3.36 eV), ITO/ZnO NPs/PEI (5 nm) (3.75 eV), ITO/ZnO NPs/Arg (5 nm) (3.56 eV), ITO/ZnO NPs/Arg (10 nm) (3.67 eV), ITO/ZnO NPs/Arg (20 nm) (3.84 eV), and ITO/ZnO NPs/Arg (30 nm) (3.53 eV) eV). In Figure 3c, the electronic energy levels are illustrated, demonstrating shifts in work functions upon the insertion of ILs such as Arg and PEI between the ZnO NP EIL and QD EML. The work functions of all samples containing Arg (5 nm, 10 nm, 20 nm, and 30 nm) and PEI ILs showed an increase compared to that of ITO/ZnO NPs. This indicates a rise in energy barriers for electron injection. When the PEI (5 nm) IL was coated onto the ZnO NPs, the work function increased more significantly compared to when the Arg IL (5 nm) was applied. The work function consistently increased with the increasing thickness of the Arg IL, except in the case where the thickness reached 30 nm. This trend suggests a potential decrease in the current density of the IQLED as the thickness of the Arg IL increased.

To enhance our comprehension of the molecular interaction between ZnO NPs and ILs (Arg and PEI), XPS measurements were performed. This analysis focused on the intensity of an interfacial dipole linked to protonated amines within a molecular layer, such as Arg and PEI [31]. The objective was to investigate the influence of PEI (5 nm) and varying thicknesses of Arg (0 nm, 5 nm, 10 nm, 20 nm, and 30 nm) on ITO/ZnO NPs substrates using XPS analysis.

Figure 4a shows Zn 2p peaks at 1043.97 eV and 1021.26 eV for ITO/ZnO NPs, 1043.97 eV and 1021.02 eV for ITO/ZnO NPs/PEI (5 nm), 1043.97 eV and 1021.75 eV for ITO/ZnO NPs/Arg (5 nm), 1043.97 eV and 1021.53 eV for ITO/ZnO NPs/Arg (10 nm), 1043.97 eV and 1021.57 eV for ITO/ZnO NPs/Arg (20 nm), and 1043.97 eV and 1021.41 eV for ITO/ZnO NPs/Arg (30 nm). The energy differences of 22.71 eV for ITO/ZnO NPs, 22.95 eV for ITO/ZnO NPs/PEI (5 nm), 22.22 eV for ITO/ZnO NPs/Arg (5 nm), 22.44 eV for ITO/ZnO NPs/Arg (10 nm), 22.40 eV for ITO/ZnO NPs/Arg (20 nm), and 22.56 eV for ITO/ZnO NPs/Arg (30 nm) between two peaks indicate a normal chemical state of Zn^2+^ for the compounds [36,39]. In comparison with the ZnO NPs, the binding energy location for the PEI-modified ZnO NPs slightly decreased, but Arg-modified ZnO increased, indicating that the electronic structure of ZnO is influenced by the formation of the Zn-N chemical bonding [36]. 

Figure 4b shows that the N 1s peaks in the films consisted of ITO/ZnO NPs/PEI (5 nm), ITO/ZnO NPs/Arg (5 nm), ITO/ZnO NPs/Arg (10 nm), ITO/ZnO NPs/Arg (20 nm), and ITO/ZnO NPs/Arg (30 nm) thin films. The thin films exhibited an asymmetric N 1s spectra. The spectra were deconvoluted into three binding states, attributed to nitrogen bonded to Zn in the metal oxide lattice, the nitrogen atoms in the neutral amines, and the nitrogen atoms in protonated amines. The high resolution XPS spectra of N 1s exhibited three asymmetric peaks at 399.25 eV, 399.98 eV, and 400.79 eV for ITO/ZnO NPs/PEI (5 nm) film; at 398.98 eV, 399.98 eV, and 400.79 eV for ITO/ZnO NPs/Arg (5 nm) film; at 399.46 eV, 399.91 eV, and 400.13 eV for ITO/ZnO NPs/Arg (10 nm) film; at 399.21 eV, 399.95 eV, and 400.39 eV for ITO/ZnO NPs/Arg (20 nm) film; and at 399.14 eV, 399.77 eV, and 400.12 eV for ITO/ZnO NPs/Arg (30 nm) film, as shown in Figure 4b. The presence of protonated amines, known for inducing electrostatic dipoles [33,40], is responsible for the intensity of an interfacial dipole. In all samples coated with Arg and PEI ILs, distinct features were observed: C 1s peaks near 286 eV and nitrogen 1s peaks correspond to protonated amines (slightly higher than 400 eV) and nonprotonated amines (slightly less than 400 eV). The concentration of protonated nitrogen (N^+^) in the following structures—ITO/ZnO NPs/PEI (5 nm), ITO/ZnO NPs/Arg (5 nm), ITO/ZnO NPs/Arg (10 nm), ITO/ZnO NPs/Arg (20 nm), and ITO/ZnO NPs/Arg (30 nm)—were calculated as 2.27%, 3.76% 10.88%, 12.91%, and 18.20%, respectively. Additionally, the O 1s peaks for thin films of ITO/ZnO NPs and their respective combinations—ITO/ZnO NPs/PEI (5 nm), ITO/ZnO NPs/Arg (5 nm), ITO/ZnO NPs/Arg (10 nm), ITO/ZnO NPs/Arg (20 nm), and ITO/ZnO NPs/Arg (30 nm)—were estimated to be 530.26 eV, 530.99 eV, 530.55 eV, 530.53 eV, 530.87 eV, and 530.91 eV, respectively, as shown in Figure 4c. When utilizing the ILs, a reduction in the hydroxyl group ratio occurs due to the substitution of these groups in Arg. It appears that the amine groups in Arg replaced more hydroxyl groups compared to those in PEI. Furthermore, with increasing Arg thickness, there was a decrease in the hydroxyl group ratio and an increase in the carboxyl group ratio, as shown in Figure 4c. The carboxyl group has the ability to donate a proton (H^+^) in a chemical reaction, while the amine group, containing nitrogen, acts as a base by accepting protons. The protonated amine groups align in proximity to the surface of ZnO NPs, whereas the negatively charged carboxyl groups are externally positioned on the Arg IL. This configuration gives rise to the creation of electrostatic dipoles [41,42]. When these dipoles are aligned perpendicularly to the surface and directed inward, their formation results in an increase in the work function, illustrated in Figure 4d. This correlation was confirmed by the UPS results. Figure 4d presents a physical model proposed based on the properties of Arg. 

To investigate the impact of Arg or PEI ILs on exciton quenching occurring on the ZnO NP surface, we conducted PL and TRPL analyses for QD, ZnO NPs/QD, ZnO NPs PEI (5 nm)/QD, and ZnO NPs/Arg (20 nm)/QD. Various samples were prepared, including glass/QD, glass/ZnO NPs/QD, glass/ZnO NPs/PEI (5 nm)/QD, and glass/ZnO NPs/Arg (20 nm)/QD. Figure 5a illustrates the PL spectra of all four samples. The ZnO NPs/QD system exhibits the lowest PL intensity, indicating the strong luminescence quenching nature of the ZnO NPs, which is attributed to their metallic nature. It was demonstrated that ILs such as PEI or Arg effectively suppressed exciton quenching compared to the ZnO NPs/QD system without the ILs. Figure 5b shows that PL decay properties were recorded to examine the interaction between ZnO NPs, ZnO NPs/PEI (5 nm), and ZnO NPs/Arg (20 nm) EILs and QD EMLs. The PL lifetimes for QD, ZnO NPs/QD, ZnO NPs PEI (5 nm), and ZnO NPs/Arg (20 nm) were recorded to be 4.54 ns, 4.02 ns, 5.17 ns, and 4.75 ns, respectively. This indicates that the PL lifetime of the samples with Arg or PEI IL increased compared to the ZnO NPs/QD sample. It suggests that Arg or PEI IL suppresses exciton quenching. 

With the increased energy barriers for electron injection in IQLEDs with PEI and the Arg ILs on the ZnO NP EIL, it is expected that the current density for IQLEDs should decrease. To compare the effect of the Arg and PEI ILs, we fabricated IQLEDs using ZnO NPs (50 nm), ZnO NPs (50 nm)/PEI (5 nm), and ZnO NPs (50 nm)/Arg (20 nm) EILs. The EL characteristics of these IQLEDs are illustrated in Figure 6. Our findings revealed a decrease in current density upon integration of the Arg and PEI ILs on ZnO NPs, indicating reduced electron injection compared to IQLEDs with ZnO NPs. It is believed that the decrease in current density is attributed to the increased energy barriers for electron injection, combined with the inherent insulating nature of the ILs. As illustrated in Figure 6a, the IQLED with a 20 nm-thick Arg IL demonstrated a more substantial reduction in current density compared to the IQLED with a 5 nm-thick PEI IL. The luminance values of IQLEDs followed similar trends as the current density. While the maximum current efficiencies of IQLEDs with ZnO NPs, ZnO NPs/PEI, and ZnO NPs/Arg EILs were estimated to be 16.51 cd/A, 27.33 cd/A, and 36.67 cd/A, respectively, the EQEs of these IQLEDs were determined to be 3.91%, 6.56%, and 8.93%, respectively. The IQLED with ZnO NPs/Arg (20 nm) EIL exhibited a 2.22-fold higher current efficiency and a 2.28-fold higher EQE compared to the IQLED with ZnO NPs EIL, and it demonstrated a 1.34-fold higher current efficiency and a 1.36-fold higher EQE compared to the IQLED with ZnO NPs/PEI (5 nm) EIL. Despite having the lowest current density, the IQLED with Arg (20 nm) IL exhibited the highest maximum current efficiency and EQE. This could be attributed to the optimal charge balance achieved in the QD EML and d minimized exciton quenching according to the low concentration of hydroxyl groups, as confirmed by XPS analysis, thereby enhancing luminance [2,31,43]. The key parameters of IQLEDs with ZnO NPs, ZnO NPs/PEI (5 nm), and ZnO NPs/Arg (20 nm) are summarized in Table 1.

To evaluate the effects of varying Arg IL thicknesses, IQLEDs were fabricated using Arg ILs of varying thicknesses (5 nm, 10 nm, 20 nm, and 30 nm). In Figure 7, the EL characteristics of these IQLEDs are illustrated. In Figure 7a, the plotted current density curves against applied voltage clearly depict a trend that as the thickness of the Arg IL increased, the current density of the IQLEDs decreased. Additionally, in Figure 7b, the maximum luminance values of IQLEDs with varying Arg IL thicknesses were estimated as follows: 158,567.8 cd/m^2^ for the IQLED with a 5 nm-thick Arg IL, 197,017.8 cd/m^2^ for the IQLED with a 10 nm-thick Arg IL, 217,780.8 cd/m^2^ for the IQLED with a 20 nm-thick Arg IL, and 57,228.9 cd/m^2^ for the IQLED with a 30 nm-thick Arg IL. Among these, the IQLED with a 20 nm-thick Arg IL exhibited the highest luminance, indicating the optimal charge balance within the QD EML. Figure 7c illustrates the maximum current efficiencies of IQLEDs with Arg ILs with thicknesses of 5 nm, 10 nm, 20 nm, and 30 nm, measuring at 19.53 cd/A, 21.89 cd/A, 36.67 cd/A, and 27.35 cd/A, respectively. Similarly, in Figure 7d, the maximum EQEs for these IQLEDs with 5 nm-thick, 10 nm-thick, 20 nm-thick, and 30 nm-thick Arg ILs were calculated at 4.7%, 5.27%, 8.93%, and 6.61%, respectively. We believe that current efficiency and EQE are influenced by the charge balance within the QD and the exciton quenching at the interface between the QD EML and Arg IL. As a result, the IQLED with a 20 nm-thick Arg IL exhibits a 2.28-fold improvement in EQE compared to the IQLED without an IL, while the IQLED with a 20 nm-thick Arg IL shows a 1.36-fold improvement in EQE compared to the IQLED with a 5 nm-thick PEI IL. Table 2 summarizes the key parameters of IQLEDs with ZnO NPs/Arg (5 nm), ZnO NPs/Arg (10 nm), ZnO NPs/Arg (20 nm), and ZnO NPs/Arg (30 nm) EILs.

To further explore the effect of Arg on electron injection, we fabricated the electron-only devices (EODs) and hole-only device (HOD). We developed six EOD variants with different Arg thicknesses (0 nm, 5 nm, 10 nm, 20 nm, and 30 nm) and PEI (5 nm). The structures of the EODs consisted of ITO (150 nm)/ZnO NPs (50 nm)/Arg (0 nm, 5 nm, 10 nm, 20 nm, and 30 nm) and PEI (5 nm)/QD (10 nm)/Ag (100 nm). The structure of the HOD comprised ITO (150 nm)/QD (10 nm)/TCTA (40 nm)/WO_x_ (10 nm)/Ag (100 nm). Electron injection occurred from the cathode to the conduction band minimum (CBM) level of QDs via the CBM of ZnO NPs/Arg EILs. Upon comparing the current densities of the EODs using Arg ILs with varying thicknesses to those of EODs with ZnO NPs and ZnO NPs/PEI (5 nm) ILs, in Figure 8, it was observed that, except for the EOD with a 30 nm-thick Arg IL, the energy barriers for electron injection in EODs with varying thicknesses of Arg ILs were higher than those in EODs with ZnO NPs and ZnO NPs/PEI EILs. This suggests that the insulating nature of Arg surpasses that of PEI. An EOD with a 30 nm-thick Arg exhibited a lower barrier for electron injection compared to the EOD with a 10 nm-thick Arg, indicating that the EOD with a 30 nm-thick Arg is more influenced by its inherent insulating nature than the effect of the energy barrier. The current density of the HOD was observed to be significantly smaller than the EOD current densities, suggesting that hole injection is much more challenging than electron injection.

Surface morphology properties of the ITO/ZnO NPs and ITO/ZnO NPs/Arg ILs (5 nm, 10 nm, 20 nm, and 30 nm) were characterized via AFM measurements, and the results are presented in Figure 9. The ITO/ZnO NP film exhibited the largest roughness, with a root-mean-square (RMS) value of 0.600 nm. Upon the application of Arg and PEI ILs on the ZnO NP surface, smoother surfaces were observed, indicating complete coverage by Arg and PEI ILs. This suggests that as the thickness of the Arg increased, the reduced surface roughness was attributed to the replacement of more hydroxyl groups on the ZnO NPs with more amine groups in the Arg molecule. Interestingly, applying Arg on the ZnO NP surface resulted in a rougher surface compared to when PEI was used. This might be attributed to Arg molecules, which align perpendicularly to the ZnO NP surface, contrasting with the horizontal application of PEI, as reported by Li et al. [36]. 

Figure 10 shows the normalized electroluminescence spectra of IQLEDs with ZnO NPs, ZnO NPs/PEI (5 nm), ZnO NPs/Arg (5 nm), ZnO NPs/Arg (10 nm), ZnO NPs/Arg (20 nm), and ZnO NPs/Arg (30 nm) at the applied voltage for maximum luminescence. The full width at half maximum (FWHM) values of the IQLEDs with ZnO NPs, ZnO NPs/PEI (5 nm), ZnO NPs/Arg (5 nm), ZnO NPs/Arg (10 nm), ZnO NPs/Arg (20 nm), and ZnO NPs/Arg (30 nm) at the point of maximum luminance were estimated to be 38.99 nm, 38.81 nm, 38.19 nm, 38.19 nm, 38.49 nm, and 39.41 nm, respectively. The comprehensive characteristic parameters are summarized in Table 3. 

Appendix A illustrates the EL spectra of IQLEDs with ZnO NPs and ZnO NPs/Arg (20 nm) EILs at different applied voltages. It is evident that the EL spectra of the IQLED with a 20 nm-thick Arg IL exhibits good monochromaticity, with no shift in the light-emitting peak position, as shown in Appendix A. This indicates a high level of carrier transmission, demonstrating good charge balance and preventing exciton diffusion to an adjacent luminescent layer. Conversely, the EL spectra of the IQLED without a 20 nm-thick Arg IL show a red shift with increasing voltage, and the Stark effect under the electric field is notably pronounced, as shown in Appendix A. Appendix A displays the FWHM values and peak positions of electroluminescence spectra of IQLEDs, with (a) ZnO NPs EIL and (b) ZnO NPs/Arg (20 nm) EIL at different applied voltages.

## 4. Conclusions

We demonstrated the highly effective role of Arg on the ZnO NP EIL as an IL in an IQLED, significantly suppressing electron injection from the ZnO NP EIL into the QD EML, thereby enhancing device performance. UPS analysis revealed that the ZnO NPs/Arg (20 nm) IL exhibited the highest work function of 3.84 eV compared to 3.36 eV for ZnO NPs and 3.75 eV for ZnO NPs/PEI (5 nm). As a result, the Arg IL (20 nm) notably increased the energy barrier for electron injection from the ZnO NP EIL into the QD EML. The introduction of Arg lL on the ZnO NP EIL effectively increased the energy barrier, substantially reducing electron injection into the QD EML. XPS analysis revealed that the amine group in Arg replaced the hydroxyl group on ZnO NPs. We proposed a physical model where carboxyl negative ions (-COO^−^) were located at one end of the molecule, while the amine positive ions (-NH_3_^+^) were located at the other end. This configuration resulted in the creation of electrostatic dipoles, which are anticipated to increase the energy barrier for electron injection. EOD analysis indicated that the Arg IL not only increased the energy barrier for electron injection, but also exhibited an inherent insulation nature. Further AFM analysis indicated that the Arg IL aligns perpendicularly to the ZnO NP surface, contrasting with the horizontal application of PEI. The IQLED with Arg (20 nm) EIL exhibited a 2.22-fold increase in current efficiency and a 2.28-fold rise in EQE compared to the IQLED with ZnO NP EIL. Additionally, it demonstrated a 1.34-fold increase in current efficiency and a 1.36-fold rise in EQE compared to the IQLED with ZnO NPs/PEI (5 nm) EIL. The remarkable enhancement in device performance can be attributed to the following: (1) the reduced electron injection of the Arg IL; (2) the suppression of exciton quenching and the nonradiative recombination process at the interface of the ZnO NP EIL with Arg IL and QD EML; (3) the balanced electron–hole recombination. 

## Figures and Tables

**Figure 1 nanomaterials-14-00266-f001:**
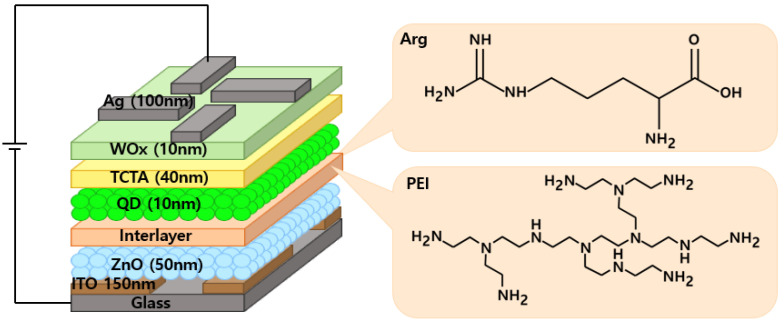
Structure of IQLEDs using Arg and PEI interlayers on ZnO NP layer and chemical structures of Arg and PEI.

**Figure 2 nanomaterials-14-00266-f002:**
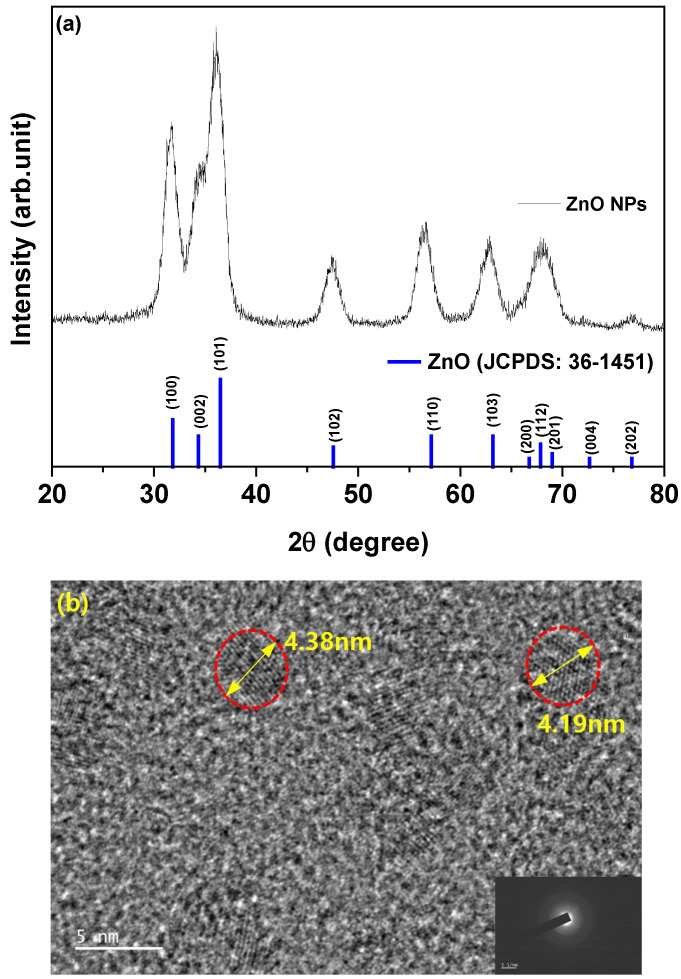
(**a**) X-ray diffraction patterns of ZnO nanoparticles over a 2θ range of 20–80° and (**b**) field-emission transmission electron microscopy of ZnO nanoparticles.

**Figure 3 nanomaterials-14-00266-f003:**
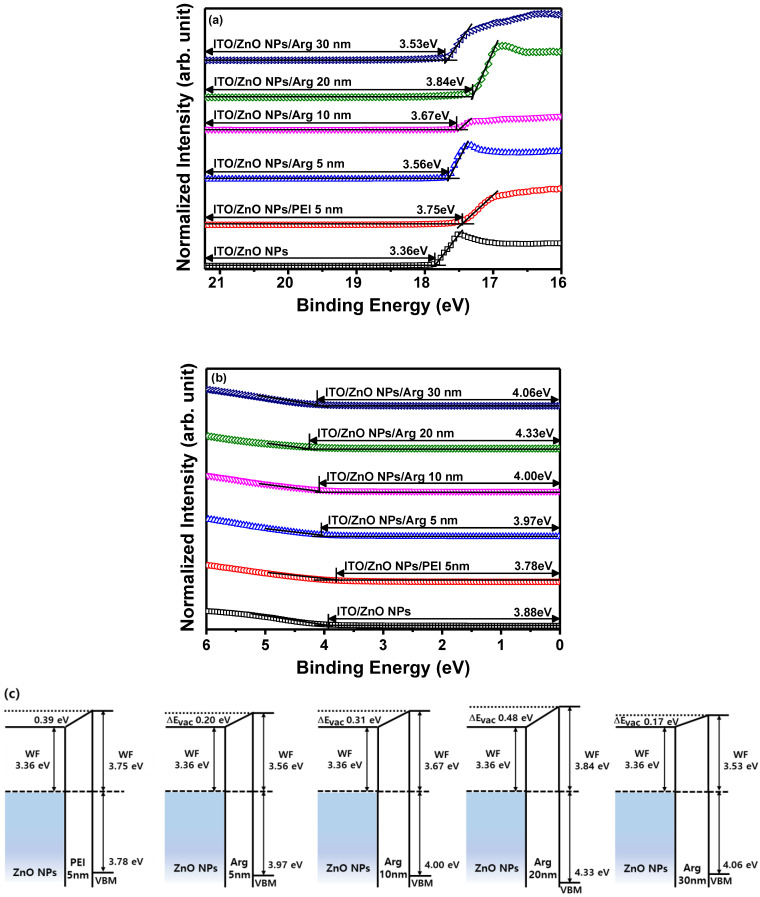
(**a**) Ultraviolet photoelectron spectroscopy spectra of secondary electron cutoff regions and (**b**) valence band regions of ITO/ZnO NPs, ITO/ZnO NPs/PEI (5 nm), ITO/ZnO NPs/Arg (5 nm), ITO/ZnO NPs/Arg (10 nm), ITO/ZnO NPs/Arg (20 nm), and ITO/ZnO NPs/Arg (30 nm). (**c**) Schematic energy level alignment diagrams of ZnO NPs/PEI (5 nm), ZnO NPs/Arg (5 nm), ZnO NPs/Arg (10 nm), ZnO NPs/Arg (20 nm), and ZnO NPs/Arg (30 nm).

**Figure 4 nanomaterials-14-00266-f004:**
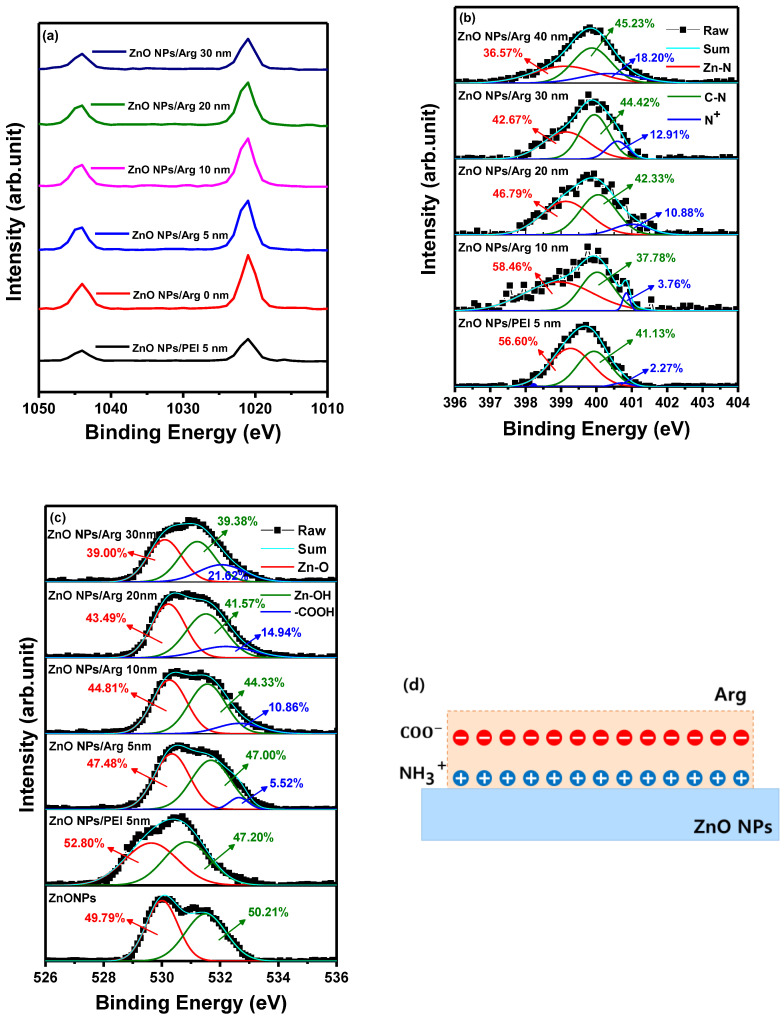
X-ray photoelectron spectroscopy spectra of ITO/ZnO NPs, ITO/ZnO NPs/PEI (5 nm), ITO/ZnO NPs/Arg (5 nm), ITO/ZnO NPs/Arg (10 nm), ITO/ZnO NPs/Arg (20 nm), and ITO/ZnO NPs/Arg (30 nm): (**a**) Zn 2p, (**b**) N 1s, and (**c**) O 1s. (**d**) Schematic of proposed physical model for creating an electrostatic dipole in Arg layer.

**Figure 5 nanomaterials-14-00266-f005:**
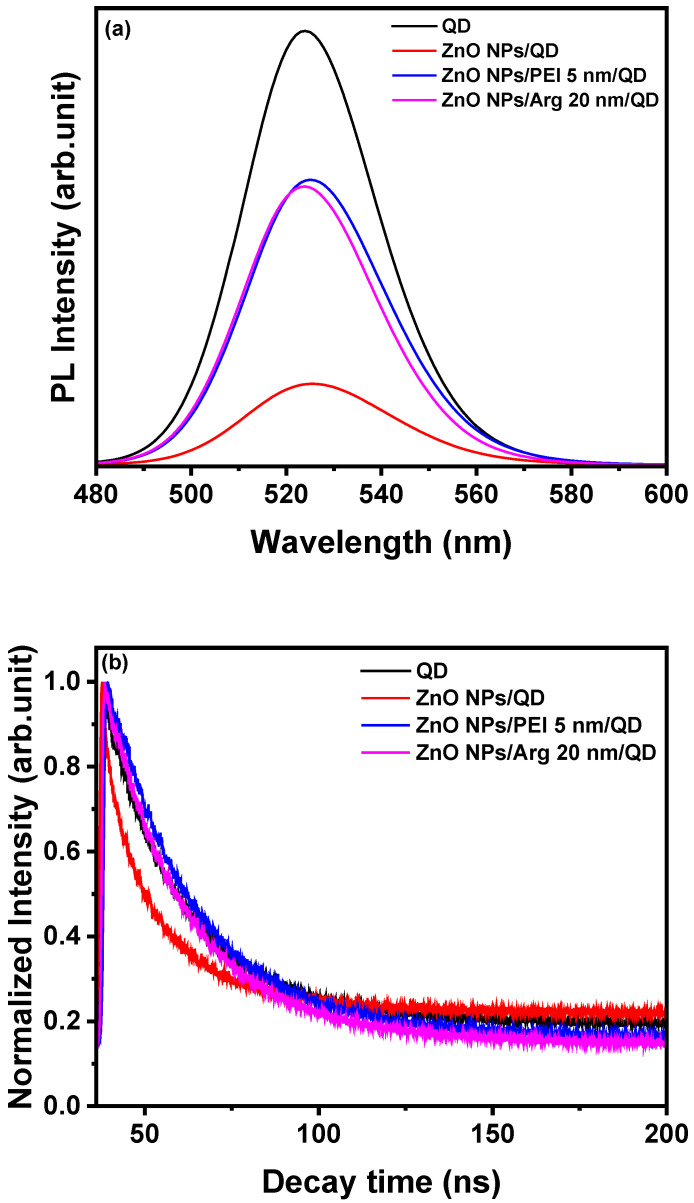
(**a**) Photoluminescence of QD, ZnO NPs/QD, ZnO NPs/PEI (5 nm)/QD, and ZnO NPs/Arg (20 nm)/QD. (**b**) Time-resolved photoluminescence of QD, ZnO NPs/QD, ZnO NPs/PEI (5 nm)/QD, and ZnO NPs/Arg (20 nm)/QD.

**Figure 6 nanomaterials-14-00266-f006:**
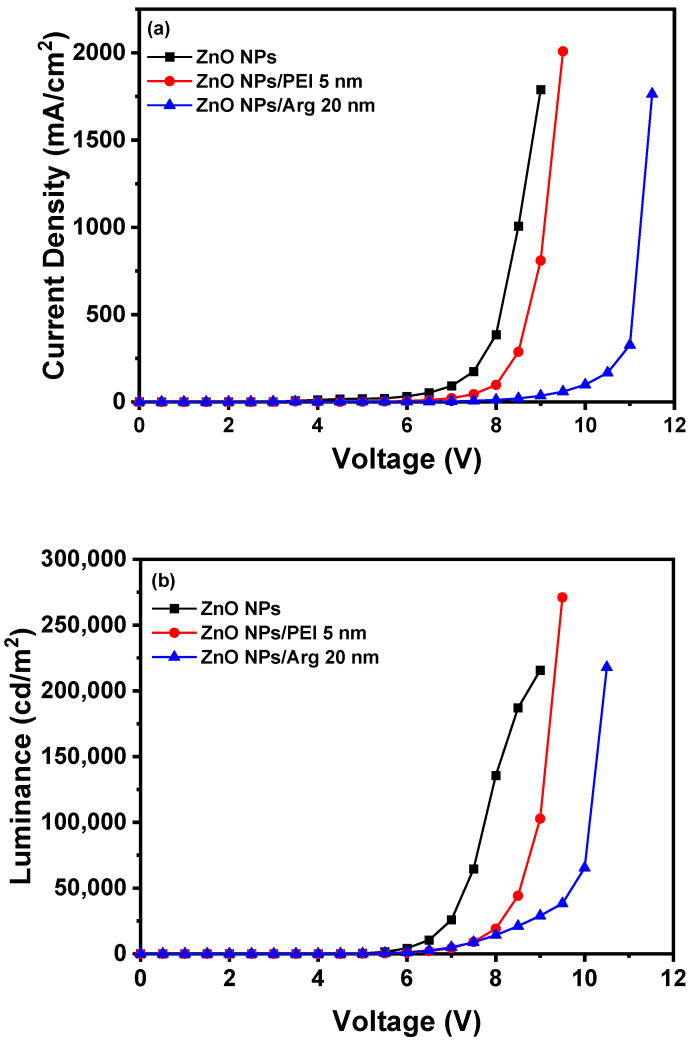
Electroluminescence characteristics of IQLEDs with ZnO NPs, ZnO NPs/Arg (20 nm), and ZnO NPs/PEI (5 nm) electron injection layers: (**a**) current density–voltage curves, (**b**) luminance–voltage curves, (**c**) current efficiency–luminance curves, and (**d**) external quantum efficiency–luminance curves.

**Figure 7 nanomaterials-14-00266-f007:**
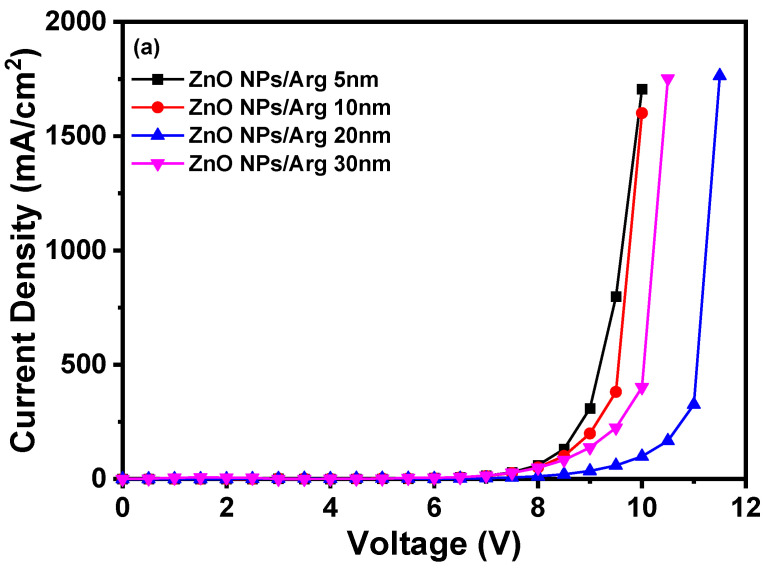
Electroluminescence characteristics of IQLEDs with ZnO NPs/Arg (5 nm), ZnO NPs/Arg (10 nm), ZnO NPs/Arg (20 nm), and ZnO NPs/Arg (30 nm): (**a**) current density–voltage curves, (**b**) luminance–voltage curves, (**c**) current efficiency–luminance curves, and (**d**) external quantum efficiency–luminance curves.

**Figure 8 nanomaterials-14-00266-f008:**
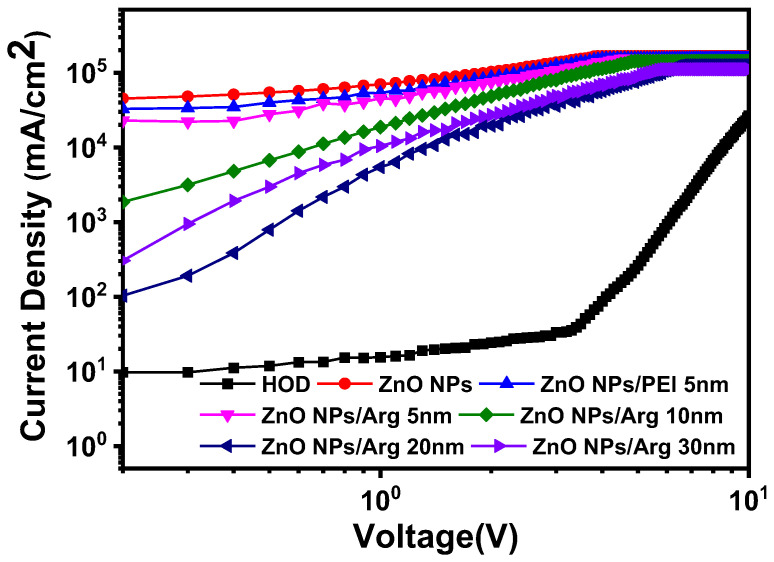
Comparison of the current density curves as an applied voltage for electron-only devices, ITO (150 nm)/ZnO NPs (50 nm)/Arg (0 nm, 5 nm, 10 nm, 20 nm, and 30 nm) and PEI (5 nm)/QD (10 nm)/Ag (100 nm), and hole-only device, ITO (150 nm)/QD (10 nm)/TCTA (40 nm)/WO_x_ (10 nm)/Ag (100 nm).

**Figure 9 nanomaterials-14-00266-f009:**
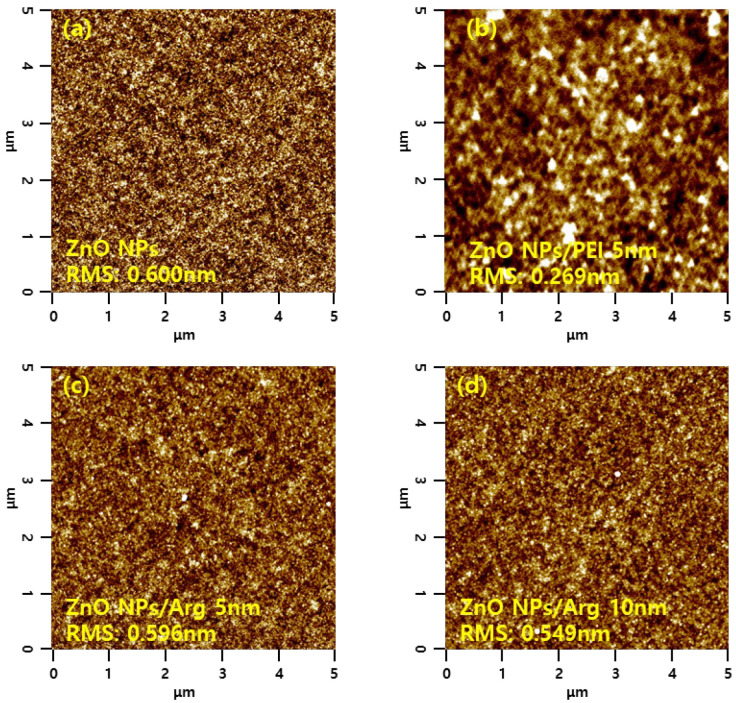
Atomic force microscope images of (**a**) ZnO NPs, (**b**) ZnO NPs/PEI (5 nm), (**c**) ZnO NPs/Arg (5 nm), (**d**) ZnO NPs/Arg (10 nm), (**e**) ZnO NPs/Arg (20 nm), and (**f**) ZnO NPs/Arg (30 nm) films.

**Figure 10 nanomaterials-14-00266-f010:**
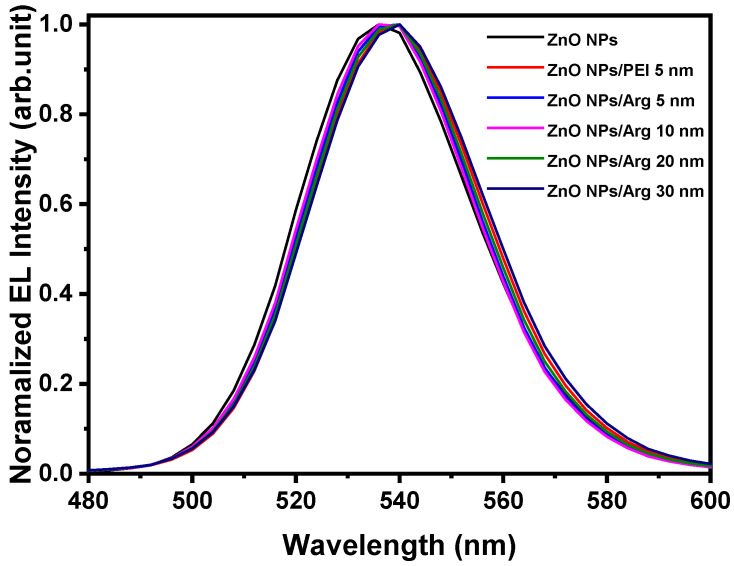
Normalized electroluminescence spectra of IQLEDs with ZnO NPs, ZnO NPs/PEI (5 nm), ZnO NPs/Arg (5 nm), ZnO NPs/Arg (10 nm), ZnO NPs/Arg (20 nm), and ZnO NPs/Arg (30 nm) at the applied voltage for maximum luminescence.

**Table 1 nanomaterials-14-00266-t001:** The performance parameters of IQLEDs with ZnO NPs, ZnO NPs/PEI (5 nm), and ZnO NPs/Arg (20 nm) EILs.

EILs	Turn on Voltage at 1 cd/m^2^ (V)	Current Density at 9 V (mA/cm^2^)	Maximum Luminance (cd/m^2^)	Maximum Current Efficiency (cd/A)	Maximum EQE (%)
ZnO NPs	4.50	1788.6	215,627.6	16.51	3.91
ZnO NPs/PEI	4.52	809.85	270,995.6	27.87	6.70
ZnO NPs/Arg	4.01	35.28	217,780.8	36.67	8.93

**Table 2 nanomaterials-14-00266-t002:** The performance parameters of IQLEDs with ZnO NPs/Arg (5 nm), ZnO NPs/Arg (10 nm), ZnO NPs/Arg (20 nm), and ZnO NPs/Arg (30 nm) EILs.

Thickness of ZnO/Arg EIL (nm)	Turn-on Voltage at 1 cd/m^2^ (V)	Current Density at 9 V (mA/cm^2^)	Maximum Luminance (cd/m^2^)	Maximum Current Efficiency (cd/A)	Maximum EQE (%)
5	4.02	309.31	158,567.8	19.53	4.69
10	4.02	199.15	197,017.8	21.89	5.26
20	4.01	35.28	217,780.8	36.67	8.92
30	4.00	137.64	57,228.9	27.35	6.60

**Table 3 nanomaterials-14-00266-t003:** Normalized electroluminescence spectra of IQLEDs with ZnO NPs, ZnO NPs/PEI (5 nm)/ZnO NPs/Arg (5 nm)/ZnO NPs/Arg (10 nm), ZnO NPs/Arg (20 nm), and ZnO NPs/Arg (30 nm) EILs.

EIL	EL Wavelength (nm)	FWHM (nm)
ZnO NPs	536	38.99
ZnO NPs/PEI 5 nm	540	38.81
ZnO NPs/Arg 5 nm	540	38.19
ZnO NPs/Arg 10 nm	536	38.17
ZnO NPs/Arg 20 nm	540	38.49
ZnO NPs/Arg 30 nm	540	39.41

## Data Availability

The data that support the findings of this study are available from the corresponding authors upon reasonable request.

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
