# Peer review of "Enhancing Efficiency in Inverted Quantum Dot Light-Emitting Diodes through Arginine-Modified ZnO Nanoparticle Electron Injection Layer"

_nanomaterials, 2024, doi:10.3390/nano14030266_

Round 1

Reviewer 1 Report

Comments and Suggestions for Authors

In this paper, the authors investigated the effect of the modified layer on the device performance by using Arginine Modified ZnO Nanoparticle as Electron Injection Layer. The authors should explain or modify the following questions:

1. It is mentioned in the text that polyethyleneimine (PEI) increases the work function, but the effect of the intrinsic molecular dipole of PEI associated with the interfacial dipole on decreasing the work function of ZnO has been widely reported. Why is it contrary to the results reported in the relevant literature? 1. (ACS Appl. Mater. Interfaces 2017, 9, 20231−20238) 2. (Adv. Funct. Mater.2014, 24, 3808–3814) 3. (Nanoscale, 2018, 10, 2623–2631)

2. There is an error in the labeling of c, and d in Figure 4.

3. In figure 4c, the three peaks of fitted O 1s characteristic peak should be mentioned. this work illustrates the substitution of the -OH group on the surface of ZnO, but the control of Arg has only 5 nm PEI, and the difference between 5 nm Arg and 5 nm PEI is 0.2%, which does not fully justify the conclusion. The comparison between the 20 nm Arg-device with the best PEI or with the 20 nm PEI-device should be provided.

4. In this paper, the devices with 5 nm, 10 nm, 20 nm and 30 nm Arg as modification layer were compared and it was concluded that 20 nm is the best, but the reference one is only the device with PEI thickness of 5 nm, different thicknesses PEI modified device performance should be provided.

5. Please clarify why the device efficiency roll-off of 5 nm PEI-device is much lower than that for 20 nm Arg-device (Figure 5c and d).

6. Please use spaces between the names and units of the horizontal coordinates in Figures 5 and 6, and lowercase the first letter of the unit of luminance.

7. In the region of high driving voltage in Figure 6b, at the same luminance, for example, 50000 cd/m2, the driving voltage increases with the increase of Arg thickness, so why does the brightening voltage in Table 2 decrease with the increase of arg thickness instead?

8. In figure 8, the normalized EL spectra intensity have all been normalized and the intensity should be the same, there is an error here.

9. The formatting of some of the units in the article is problematic, so please check it thoroughly, and there are no spaces in the numbers and units.

Author Response

Response to Reviewers

The authors are grateful for the valuable comments and suggestions from the reported reviewers about our manuscript (manuscript ID: nanomaterials-2820091). We have addressed the comments raised by the reviewers, and the amendments are highlighted in yellow in the revised manuscript. The English in the manuscript has been professionally revised by a professional native speaker. Our detailed responses to the reviewers’ comments are as follows. 

Reviewer 1

In this paper, the authors investigated the effect of the modified layer on the device performance by using Arginine Modified ZnO Nanoparticle as Electron Injection Layer. The authors should explain or modify the following questions:

Q1) It is mentioned in the text that polyethyleneimine (PEI) increases the work function, but the effect of the intrinsic molecular dipole of PEI associated with the interfacial dipole on decreasing the work function of ZnO has been widely reported. Why is it contrary to the results reported in the relevant literature? 1. (ACS Appl. Mater. Interfaces 2017, 9, 20231−20238) 2. (Adv. Funct. Mater.2014, 24, 3808–3814) 3. (Nanoscale, 2018, 10, 2623–2631)

Response 1) We are very grateful for the professional opinions of the reviewer. Authors were aware of multiple studies suggesting that the PEI layer reduces the work function of ZnO. We have spent considerable time contemplating the disparities between our experimental results and the findings reported in the existing literature. Despite conflicting results, UPS results performed at two separate institutions (Korea Basic Science Institute, Joenju, South Korea and Clare Science Research Institute, Cheonan, South Korea) exhibited consistent patterns. This was reinforced by current density measurements corresponding to applied voltage, prompting the development of this paper.

Q2) There is an error in the labeling of c, and d in Figure 4.

Response 2) Thank you very much for your advice. We are very sorry for the error. We have made correction in the revised manuscript.

Q3) In figure 4c, the three peaks of fitted O 1s characteristic peak should be mentioned. this work illustrates the substitution of the -OH group on the surface of ZnO, but the control of Arg has only 5 nm PEI, and the difference between 5 nm Arg and 5 nm PEI is 0.2%, which does not fully justify the conclusion. The comparison between the 20 nm Arg-device with the best PEI or with the 20 nm PEI -device should be provided.

Response 3) We deeply appreciate the valuable insights provided by the reviewer. We investigated the characteristics of QLEDs with PEI ILs at thicknesses of 3 nm, 5 nm, 7 nm, and 10 nm, which are newly added in Supporting Information. The results confirmed that the maximum efficiency was achieved with a 5-nm-thick PEI IL. As outlined in the manuscript, we further investigated the characteristics of QLEDs with Arg ILs at thicknesses of 5 nm, 10 nm, 20 nm, and 30 nm. Notably, QLEDs incorporating a 20 nm-thick Arg IL demonstrated the highest efficiency. Subsequently, we examined the characteristics by fabricating QLEDs with the most efficient 5 nm-thick PEI IL among PEI ILs and 20 nm-thick Arg IL among Arg ILs, comparing their characteristics. The difference in -OH group concentration obtained from the 5 nm PEI and 20 nm Arg samples is 5.63%. These results were also confirmed by newly conducted experiments on photoluminescence and TRPL added in the revised manuscript.

Q4) In this paper, the devices with 5 nm, 10 nm, 20 nm and 30 nm Arg as modification layer were compared and it was concluded that 20 nm is the best, but the reference one is only the device with PEI thickness of 5 nm, different thicknesses PEI modified device performance should be provided.

Response 4) We thank the reviewer’s valuable suggestion. Upon reviewer’s request, we added electroluminescence characteristics of QLEDs with 3-nm-thick, 5-nm-thick, 7-nm-thick, and 10-nm-thick PEI ILs in the Supporting Information. The experimental results showed that IQLED with a 5-nm-thick PEI interface layer exhibited the highest efficiency.

Q5) Please clarify why the device efficiency roll-off of 5 nm PEI-device is much lower than that for 20 nm Arg-device (Figure 5c and d).

Response 5) We are very grateful for the professional opinions of the reviewer. In QLEDs with Arg IL, the efficiency is high near the turn-on voltage but sharply decreases beyond that point. We attribute this to the effective maintenance of charge balance primarily near the low turn-on voltage, where the current density is low. However, as the current density increases, it appears that the contribution of hole current diminishes compared to the increase in electron current as shown in results in EODs and HOD characteristics. The observed decrease in charge balance is thought to be primarily due to the smaller increase in hole current compared to electron current as the current density rises.

Q6) Please use spaces between the names and units of the horizontal coordinates in Figures 5 and 6, and lowercase the first letter of the unit of luminance.

Response 6) Thank you for your kind advice. We are sorry about those errors. We have addressed all the points raised by the reviewer in the manuscript.

Q7) In the region of high driving voltage in Figure 6b, at the same luminance, for example, 50000 cd/m2, the driving voltage increases with the increase of Arg thickness, so why does the brightening voltage in Table 2 decrease with the increase of Arg thickness instead?

Response 7) We appreciate the meticulous feedback from the reviewer and sincerely apologize for our oversight. In the high driving voltage range, as the thickness increases, the energy barrier becomes larger, and the insulation properties also increase. Consequently, it becomes challenging to inject electron current, requiring higher voltage to achieve the same luminance. The observed decrease in turn-on voltage to achieve a luminance of 1 cd/m² is believed to be due to the improved charge balance between electron and hole currents when the thickness is thicker. However, in the high driving voltage range, as the thickness of Arg increases, not only does the energy barrier for electron injection increase, but the inherent insulation properties also increase. As a result, injecting electron current becomes more challenging, necessitating a higher voltage to achieve equivalent luminance. The observed reduction in turn-on voltage, aiming for a luminance of 1 cd/m2, is attributed to the enhanced charge balance between electron and hole currents associated with an increase in thickness.

Q8) In figure 8, the normalized EL spectra intensity have all been normalized and the intensity should be the same, there is an error here.

Response 8) We express our gratitude for the valuable suggestions provided by the reviewer. We have modified all EL spectra in Figure 8 to be normalized.

Q9) The formatting of some of the units in the article is problematic, so please check it thoroughly, and there are no spaces in the numbers and units.

Response 9) We express our gratitude for the reviewer's valuable suggestions and sincerely apologize for any oversights. We have made every effort to address and correct our mistakes.

Reviewer 2 Report

Comments and Suggestions for Authors

minor revision

Author Response

Response to Reviewers

The authors are grateful for the valuable comments and suggestions from the reported reviewers about our manuscript (manuscript ID: nanomaterials-2820091). We have addressed the comments raised by the reviewers, and the amendments are highlighted in yellow in the revised manuscript. The English in the manuscript has been professionally revised by a professional native speaker. Our detailed responses to the reviewers’ comments are as follows. 

Reviewer 2

The authors reported inverted quantum dot light-emitting-diodes with an intermediate layer of Arginine inserted on ZnO, which is utilized to increase the work function of ZnO to inhibit electron injection, resulting in a more balanced charge injection of the device, as well as suppression of exciton bursts at the ETL/QD interface, which ultimately improves the efficiency of the device.

Q1) The ZnO/Arg 40nm in figure 4(c) does not seem to match the figure note.

Response 1) Thank you very much for your valuable advice. We are very sorry for the error. We have made correction in the revised manuscript.

Q2) Figure 4(d) should be correctly labeled, rather than labeled as ‘c’.

Response 2) Thank you for your valuable advice. We are very sorry for the mistake. We have made correction in the revised manuscript.

Q3) In figure 8, the name units of the horizontal coordinates are all wrong.

Response 3) Thank you for your kind advice. We are sorry about those errors. We have corrected them in the manuscript.

Q4) The solution spin-coating method is not easy to accurately control the thickness of the film layer, how is the thickness of Arginine mentioned in the paper verified? Please provide the relevant description.

Response 4) We are very grateful for the professional opinions of the reviewer. We adjusted the thickness by varying the concentration of the Arg solution during the fabrication of the films using the spin-coating method. The resulting thin films were validated using an alpha-step. This procedure was repeated several times to ensure the uniform formation of thin films with a consistent thickness.

Reviewer 3 Report

Comments and Suggestions for Authors

In this paper, the authors used an Arginine (Arg) interlayer onto the ZnO NP EIL. The Arg interlayer elevated the work function of ZnO NPs, thereby suppressing electron injection into the QD, leading to improved charge balance within the QDs. Additionally, the inherent insulating nature of the Arg interlayer prevented direct contact between QDs and ZnO NPs, reducing exciton quenching and consequently improving device efficiency. The resulting optimized current efficiency and external quantum efficiency (EQE) are 36.67 cd/A, and 8.92%, respectively. However, before the acceptance for publication, there are some revisions need to be made. Some problems are as follows. Before the paper could proceed further for publication, the authors are advised to address the following issues.

1.     The essay has many grammatical errors and poor wording, and authors should improve it.

2.     There are some formatting errors, for example in the "Device Fabrication" section, some places are not subscripted, and the units of temperature(°C) are also incorrectly labelled. In addition, there are two diagrams labelled as Figure 7 need be carefully checked and corrected.

3.     When electrostatic dipoles are aligned perpendicularly to the surface and directed inward, their formation results in an increase in the work function, illustrated in Figure 4(d). Compared to the work function of ZnO NPs/Arg (5 nm), ZnO NPs/Arg (10 nm), ZnO NPs/Arg (20 nm), why does the work function of ZnO NPs/Arg (30 nm) decrease to 3.53?

4.     The remarkable enhancement in device performance can be attributed suppression of exciton quenching and nonradiative recombination process at the interface ZnO NP EIL with Arg interlayer and QD EML. No data to confirm suppression of exciton quenching, it is suggested to provide the time-resolved PL decay spectra of QDs on ITO/ZnO NPs, ITO/ZnO NPs/PEI, ITO/ZnO NPs/Arg.

5.     Interestingly, although the IQLED with a 30 nm thickness of Arg IL showed lower current density compared to the one 20 nm thickness, despite having lower energy barriers for electron injection. But, in the Figure 5(a), the current density of ZnO NPs/Arg (30 nm) is higher than that of ZnO NPs/Arg (30 nm) at the same voltage, which contradicts the description in the article. Please carefully checked and corrected.

6.     In the Figure 7, it is recommended to introduce a hole-only device, which better accounts for the carrier injection balance in the device.

7.     In the Figure 8, Why does the FWHM of the device increase by 1.3 nm as the thickness of the Arg interlayer increases from 5 nm to 30 nm?

Comments on the Quality of English Language

The essay has many grammatical errors and poor wording, and authors should improve it.

Author Response

Response to Reviewers

The authors are grateful for the valuable comments and suggestions from the reported reviewers about our manuscript (manuscript ID: nanomaterials-2820091). We have addressed the comments raised by the reviewers, and the amendments are highlighted in yellow in the revised manuscript. The English in the manuscript has been professionally revised by a professional native speaker. Our detailed responses to the reviewers’ comments are as follows. 

Reviewer 3

In this paper, the authors used an Arginine (Arg) interlayer onto the ZnO NP EIL. The Arg interlayer elevated the work function of ZnO NPs, thereby suppressing electron injection into the QD, leading to improved charge balance within the QDs. Additionally, the inherent insulating nature of the Arg interlayer prevented direct contact between QDs and ZnO NPs, reducing exciton quenching and consequently improving device efficiency. The resulting optimized current efficiency and external quantum efficiency (EQE) are 36.67 cd/A, and 8.92%, respectively. However, before the acceptance for publication, there are some revisions need to be made. Some problems are as follows. Before the paper could proceed further for publication, the authors are advised to address the following issues.

Q1. The essay has many grammatical errors and poor wording, and authors should improve it.

Response 1) Thank you for your kind advice. We have diligently reviewed and rectified the language again. The English in this manuscript has been checked by a native speaker.

Q2. There are some formatting errors, for example in the "Device Fabrication" section, some places are not subscripted, and the units of temperature(°C) are also incorrectly labelled. In addition, there are two diagrams labelled as Figure 7 need be carefully checked and corrected ℃

Response 2) Thank you for your valuable advice. We are sorry about the errors and have made correction in the revised manuscript.

Q3. When electrostatic dipoles are aligned perpendicularly to the surface and directed inward, their formation results in an increase in the work function, illustrated in Figure 4(d). Compared to the work function of ZnO NPs/Arg (5 nm), ZnO NPs/Arg (10 nm), ZnO NPs/Arg (20 nm), why does the work function of ZnO NPs/Arg (30 nm) decrease to 3.53 eV?

Response 3) We express our gratitude for the valuable suggestions provided by the reviewer. We performed UPS measurements twice to confirm the decrease in work function, and in both measurements, we verified that the work function decreased as the thickness of Arg increased to 30 nm. We believe that the augmentation of Arg thickness to 30 nm results in the prevalence of inherent insulating properties, causing a reduction in the work function due to a significant decrease in interface electrostatic dipoles.

We conducted the UPS measurements twice and in both measurements, we confirmed that the work function decreased when the thickness of Arg increased to 30 nm, we believe that when the thickness of Arg increases to 30 nm, the dominance of inherent insulating properties leads to a decrease in the work function due to the large decrease in interface electrostatic dipoles. This was reinforced by current density measurements corresponding to applied voltage.

Q4. The remarkable enhancement in device performance can be attributed suppression of exciton quenching and nonradiative recombination process at the interface ZnO NP EIL with Arg interlayer and QD EML. No data to confirm suppression of exciton quenching, it is suggested to provide the time-resolved PL decay spectra of QDs on ITO/ZnO NPs, ITO/ZnO NPs/PEI, ITO/ZnO NPs/Arg.

Response 4) We are very grateful for the professional opinions of the reviewer. Following the recommendations of the reviewers, we conducted photoluminescence and time-resolved photoluminescence measurements to explore exciton quenching and thoroughly analyzed the collected data. The analyses of both PL and TRPL indicate that Arg reduced exciton quenching at the interface between QD EML and ZnO NP EIL.

Q5. Interestingly, although the IQLED with a 30 nm thickness of Arg IL showed lower current density compared to the one 20 nm thickness, despite having lower energy barriers for electron injection. But, in the Figure 5(a), the current density of ZnO NPs/Arg (30 nm) is higher than that of ZnO NPs/Arg (30 nm) at the same voltage, which contradicts the description in the article. Please carefully checked and corrected.

Response 5) We are very grateful for the professional opinions of the reviewer. We fully agree with your opinion and please forgive us for the carelessness in this discussion. We deleted the sentence.

Q6. In the Figure 7, it is recommended to introduce a hole-only device, which better accounts for the carrier injection balance in the device.

Response 6) We appreciate the reviewer's constructive suggestions. In response, we have incorporated data on the hole-only device into the revised manuscript.

Q7. In the Figure 8, Why does the FWHM of the device increase by 1.3 nm as the thickness of the Arg interlayer increases from 5 nm to 30 nm?

Response 7) We redrawed the nomalized EL spectra of IQLEDs with ZnO NPs, ZnO NPs/PEI (5nm), ZnO NPs/Arg (5 nm), ZnO NPs/Arg (10 nm), ZnO NPs/Arg (20 nm), and ZnO NPs/Arg (30 nm) at the applied voltage for maximum luminescence. The FWHM value of QLEDs with a 5 nm-thick Arg IL differs by 1.22 nm from that of QLEDs with a 30 nm-thick Arg interlayer. We attribute this difference to the micro-cavity effect induced by the variation in thickness.

Reviewer 4 Report

Comments and Suggestions for Authors

The presented study investigates the use of ZnO nanoparticles as electron transport materials in QLEDs, with a specific focus on challenges related to charge imbalance and exciton quenching at the interface between the quantum dot emission layer and ZnO. To tackle the charge injection imbalance in inverted QLEDs (IQLEDs), the authors propose a solution by introducing an arginine (Arg) interlayer (IL) on the ZnO electron transport layer. This work introduces a promising approach to mitigating charge injection imbalance in IQLEDs. After addressing the following points through revisions, I recommend its publication in this journal.

The observed higher turn-on voltage of over 4V in the presented IQLEDs raises questions about the underlying reasons for this phenomenon. It is advisable for the authors to engage in a thorough discussion on the factors influencing the elevated turn-on voltage.

The article lacks a comprehensive comparison of the IQLED device performance with existing technologies. To enhance the reader's understanding of the progress made, the authors are encouraged to include a performance table highlighting key parameters.

The faster EQE decay and more pronounced roll-off in IQLEDs with ZnO NPs/Arg (30 nm) compared to ZnO NPs/PEI (5 nm) as brightness increases require elucidation. The authors should delve into the reasons behind this phenomenon, providing a detailed exploration of factors such as charge transport, recombination kinetics, or potential degradation mechanisms in the interlayer.

Despite the emphasis on high brightness and EQE, the article lacks discussion on the stability of IQLED devices, a crucial aspect left unaddressed. It is recommended that the authors provide insights into the stability testing conducted on the devices.

Author Response

Response to Reviewers

The authors are grateful for the valuable comments and suggestions from the reported reviewers about our manuscript (manuscript ID: nanomaterials-2820091). We have addressed the comments raised by the reviewers, and the amendments are highlighted in yellow in the revised manuscript. The English in the manuscript has been professionally revised by a professional native speaker. Our detailed responses to the reviewers’ comments are as follows. 

Reviewer 4

The presented study investigates the use of ZnO nanoparticles as electron transport materials in QLEDs, with a specific focus on challenges related to charge imbalance and exciton quenching at the interface between the quantum dot emission layer and ZnO. To tackle the charge injection imbalance in inverted QLEDs (IQLEDs), the authors propose a solution by introducing an arginine (Arg) interlayer (IL) on the ZnO electron transport layer. This work introduces a promising approach to mitigating charge injection imbalance in IQLEDs. After addressing the following points through revisions, I recommend its publication in this journal.

Q1) The observed higher turn-on voltage of over 4V in the presented IQLEDs raises questions about the underlying reasons for this phenomenon. It is advisable for the authors to engage in a thorough discussion on the factors influencing the elevated turn-on voltage.

Response 1) We are very grateful for the professional opinions of the reviewer. In this study, we introduced Arg to reduce the excess electron current and successfully achieved a decrease in electron current. However, we regret not focusing on the optimization of hole current to enhance the charge balance. The HOD results, included in Figure 8, indicate that hole injection encounters significant difficulty. We attribute the increase in turn-on voltage to the charge imbalance resulting from the inferior hole injection compared to electron injection.

Q2) The article lacks a comprehensive comparison of the IQLED device performance with existing technologies. To enhance the reader's understanding of the progress made, the authors are encouraged to include a performance table highlighting key parameters.

Response 2) We thank the reviewer’s valuable suggestion. In the Introduction section, we have enhanced the content regarding IQLED to facilitate readers' understanding of the significant progress made in IQLEDs.

Q3) The faster EQE decay and more pronounced roll-off in IQLEDs with ZnO NPs/Arg (30 nm) compared to ZnO NPs/PEI (5 nm) as brightness increases require elucidation. The authors should delve into the reasons behind this phenomenon, providing a detailed exploration of factors such as charge transport, recombination kinetics, or potential degradation mechanisms in the interlayer.

Response 3) We attribute the faster EQE decay and more pronounced roll-off in the IQLED with ZnO NPs/Arg (30 nm) compared to IQLED with ZnO NPs/PEI (5 nm), arising from the charge imbalance due to inferior hole injection compared to electron injection as shown in EODs and HOD in Figure 7. We believe that this phenomenon becomes more pronounced as the luminance increases, mainly due to the less efficient injection of holes compared to electrons. Therefore, we believe that the faster EQE and pronounced roll-off is due to charge imbalance in the QD EML.

Q4) Despite the emphasis on high brightness and EQE, the article lacks discussion on the stability of IQLED devices, a crucial aspect left unaddressed. It is recommended that the authors provide insights into the stability testing conducted on the devices.

Response 4) We are very grateful for the professional opinions of the reviewer. We are sorry that the authors acknowledge the significance of stability; however, due to our equipment limitations and a tight deadline, we regretfully cannot conduct experiments on stability. In our future research, we will prioritize considering stability, recognizing its importance.

Reviewer 5 Report

Comments and Suggestions for Authors

Journal: Nanomaterials

Title: Enhancing Efficiency in Inverted Quantum Dot Light-Emitting Diodes through Arginine Modified ZnO Nanoparticle Electron Injection Layer

Authors have contributed significantly in developing the inverted quantum dot LEDs utilizing the facile interfacial modification among the electron injection layer and emissive layer. The present work is interesting, and the device performance is comparatively better, but the manuscript lags with some characterization aspects and few technical discussions. Considering the above, I request authors to address the following comments, thereby suggesting the major revision for the present stage.

I suggest authors to address the following comments,

1.        Authors can state the significance of amino acid selection, and why specifically this amino acid- Arginine?

2.         Drawbacks of the inverted structure and why this structure is considered in the present study can be highlighted in the introduction part.

3.         Authors can reveal the reason for selective work function reduction with the as-obtained interfacial dopant molecule.

4.         What would be the effects of nanoparticle size? In addition, how the nanoparticle dispersion can affect the electrical properties?

5.         It is advisable to compare the photoluminescent properties among control and modified perovskite QD emissive layers.

6.         Adding the PLQY discussion and TRPL lifetime discussion can comparatively help to improve the manuscript quality.

7.         Voltage vs current density plots can be helpful in understanding the leakage characteristics as well as it can give further insights about the morphological contribution.

8.         Why the as-formed QD LEDs exhibit larger FWHM as compared to conventional Quantum dot LEDs?

9.         How about the spectral stability upon biasing to higher voltage?

10.     Ion migration characteristics and the relative operational lifetime studies can be conducted to demonstrate the real-time significance of the as-formed LEDs.

11.     I suggest authors to take immense efforts in proof checking the spectral values, and also add appropriate figure labels for the audience understanding.

12.     There are few obvious abbreviation, format and grammar issues in the manuscript.

13.     Some relative backgrounds can be enhanced, and the following papers will be commended by future readers if possibly cited:

ACS Omega 2020, 5, 15, 8972–8981. https://doi.org/10.1021/acsomega.0c00758

ACS Appl. Mater. Interfaces 2023, 15, 2, 3644–3650. https://doi.org/10.1021/acsami.2c19123

All the best for the revision. After receiving the convincing point-to-point responses, I will recommend it for the publication.

Comments on the Quality of English Language

-

Author Response

Response to Reviewer 5

The authors are grateful for the valuable comments and suggestions from the reported reviewers about our manuscript (manuscript ID: nanomaterials-2820091). We have addressed the comments raised by the reviewers, and the amendments are highlighted in yellow in the revised manuscript. The English in the manuscript has been professionally revised by a professional native speaker. Our detailed responses to the reviewers’ comments are as follows. 

Reviewer 5

Title: Enhancing Efficiency in Inverted Quantum Dot Light-Emitting Diodes through Arginine Modified ZnO Nanoparticle Electron Injection Layer

Authors have contributed significantly in developing the inverted quantum dot LEDs utilizing the facile interfacial modification among the electron injection layer and emissive layer. The present work is interesting, and the device performance is comparatively better, but the manuscript lags with some characterization aspects and few technical discussions. Considering the above, I request authors to address the following comments, thereby suggesting the major revision for the present stage.

I suggest authors to address the following comments,

Q1) Authors can state the significance of amino acid selection, and why specifically this amino acid- Arginine?

Response 1) We are very grateful for the professional opinions of the reviewer. Authors added the followings in the manuscript:

Most small molecules possess the property of forming a self-assemble monolayers (SAMs) on metal oxide surface through covalent bonding. SAMs are known to influence the work function through polar functional groups capable of playing a role in ILs [39]. Acidic groups commonly used as SAMs, such as phosphoric acid, carboxyl acid, and amino groups, necessitate the use of toxic solvents during the preparation and treatment processes. In contrast, amino acids offer inherent benefits of environment-friendly characteristics dissolved in water solution, cost-effectiveness and widespread availability [38]. Importantly, they feature hydrophilic carboxyl groups, making them soluble in water solutions, while simultaneously possessing polar amino groups that facilitate the formation of robust hydrogen bonds [37]. Consequently, amino acids can be uniformly deposited on metal oxides. We chose arginine as the interlayer in QLEDs because among amino acids, it has the highest proportion of amine groups, creating electrostatic dipoles on the ZnO surface. Additionally, the amine group is advantageous for being environmentally friendly and cost-effective.

Q2) Drawbacks of the inverted structure and why this structure is considered in the present study can be highlighted in the introduction part.

Response 2) We thank the reviewer’s valuable suggestion. In accordance with the suggestion of the reviewer, we have elaborated on the necessity and drawbacks of QLEDs with an inverted structure in the Introduction section. Furthermore, we have enhanced the content regarding IQLED to facilitate readers' understanding of the significant progress made.

Q3) Authors can reveal the reason for selective work function reduction with the as-obtained interfacial dopant molecule.

Response 3) We are very grateful for the professional opinions of the reviewer. In the manuscript, it is explained that when arginine is inserted as an interfacial layer between the QD EML and ZnO NP EIL, the protonated amine group is positioned near the surface of ZnO NP, and the negatively charged carboxyl group is located at the outer end of arginine. This arrangement, as depicted in Figure 4(d), results in the formation of an electrostatic dipole, leading to an increase in the work function.

Q4) What would be the effects of nanoparticle size? In addition, how the nanoparticle dispersion can affect the electrical properties?

Response 4) We thank the reviewer’s valuable opinions. We believe that the dispersion of the ZnO nanoparticles were dispersed uniformly. We believe it appears transparent because the dispersion was done uniformly. Holloway group at University of Florida reported light-emitting polymer/ZnO nanoparticle heterojunctions at sub-bandgap voltages. The sub-bandgap electroluminescence was attributed to an Auger-assisted energy up-conversion process at the polymer/ZnO nanoparticle interface as shown in Figure 1, the observation of which depends on strongly on the size of the nanoparticles [1]. Our group reported that OLEDs with 5-nm-thick, 10-nm-thick, and 15-nm-thick ZnO nanoparticles had luminance onset voltages of 3.1 V, 4.1 V, and 5.3 V, respectively, and 455.1 cd/m2, 194.4 cd/m2, and 34.8 cd/m2 at 8 V, respectively [2]. The OLED with the smallest ZnO nanoparticle size of 5 nm had the lowest onset voltage and highest luminance. This indicates that the Auger electron effect which induces an energy up-conversion process pccurred in the OLED with the smallest ZnO nanoparticle size.

We are sorry that authors tried to put the Figure, but we cannot put Figure 5 here. Reviewer can find Figure 1 in the reference [1].  

Figure 1. (a) Schematic energy level diagram of MEH-PPV/ZnO nanoparticle device. (b) illustration of the Auger-like energy up-conversion process at the MEH-PPV/ZnO NPs heterojunction: (1) recombination of interfacial transfer (CT) exciton, (2) resonant energy transfer between CT exciton and electron, (3) high energy Auger electron generation by energy transfer, (4) injection of the high energy electron across the MEH-PPV/ZnO nanoparticles interface, (5) radiative recombination in MEH-PPV.

References

[1] Qian et al., Nano Today, 2010, 5, 384-389.

[2] Jeong et al., Molecular Crystals and Liquid Crystals, 2018, 663, 61-70.

Q5) It is advisable to compare the photoluminescent properties among control and modified perovskite QD emissive layers.

Response 5) We thank the reiviewer’s valuable advice. As per the reviewer’s request, we have investigated the photoluminescent characteristics and incorporated them into revised manuscript.

Q6) Adding the PLQY discussion and TRPL lifetime discussion can comparatively help to improve the manuscript quality.

Response 6) We thank the reviewer’s valuable suggestion. In response to the reviewer’s request, we have examined the TRPL lifetime. However, we regret to inform you that we were unable to include the PLQY discussion due to equipment limitations and a tight deadline. We intended to analyze the PLQY in our future studies, acknowledging its significance.

Q7) Voltage vs current density plots can be helpful in understanding the leakage characteristics as well as it can give further insights about the morphological contribution.

Response 7) The table below shows the RMS values and current densities at 2 V under the leakage current range for several devices. However, no specific relationship was found between the leakage current and the roughness of the sample.

RMS (nm)

Current Density (mA/cm2)

ZnO NPs

0.600

0.05896

ZnO NPs/PEI 5 nm

0.269

0.00524

ZnO NPs/Arg 5 nm

0.596

0.00071732

ZnO NPs/Arg 10 nm

0.549

0.00653

ZnO NPs/Arg 20 nm

0.508

0.17835

ZnO NPs/Arg 30 nm

0.224

5.6761

Q8) Why the as-formed QD LEDs exhibit larger FWHM as compared to conventional Quantum dot LEDs?

Response 8) We thank the reviewer’s valuable opinion. QDs were procured from Zeus Co. Ltd. in South Korea and utilized in the study. We believe that the full width at half maximum (FWHM) value is primarily influenced by the microcavity effect.

Q9) How about the spectral stability upon biasing to higher voltage?

Response 9) We thank the reviewer’s valuable opinion. We measured the EL spectra at different applied voltages. We confirmed that the EL spectra of IQLED with a 20-nm-thick Arg IL exhibited good monochromaticity, with no shift in the light-emitting peak position. The FWHM values obtained from EL spectra measured at applied voltage of 8.5 V was observed to be slightly decreased compared to those obtained from EL spectra based on the maximum luminance. This indicates that stability is maintained even as the voltage increases.

Q10) Ion migration characteristics and the relative operational lifetime studies can be conducted to demonstrate the real-time significance of the as-formed LEDs.

Response 10) We thank reviewer’s valuable suggestion. We regret to inform you that we were unable to include the lifetime studies due to equipment limitations and a tight deadline. We are considering this as the next topic for the paper and will present it in a subsequent publication.

Q11) I suggest authors to take immense efforts in proof checking the spectral values, and also add appropriate figure labels for the audience understanding.

Response 11) We thank the reviewer’s kind advice. We have diligently reviewed and rectified them. We have made corrections.

Q12) There are few obvious abbreviation, format and grammar issues in the manuscript.

Response 12) We thank the reviewer’s kind advice. We are sorry about the errors and have made correction in the revised manuscript.

Q13) Some relative backgrounds can be enhanced, and the following papers will be commended by future readers if possibly cited:

Response 12) Thank you for recommending an excellent papers. We have included them in the manuscript.

[13] Veeramuthu, L.; Liang, E.-C.; Zhang, Z.-X.; Cho, C.-J.; Ercan, E.; Chueh, C.-C.; Chen, W.-C.; Borsali, R.; Kuo, C.-C., Improving the Performance and Stability of Perovskite Light-Emitting Diodes by a Polymer Nanothick Interlayer-Assisted Grain Control Process, Omega, 2020, 5, 8972–8981. https://doi.org/10.1021/acsomega.0c00758

[14] Zhao, Y.; Li, M.; Qin, X.; Yang, P. Zhang, W.-H.; Wei, Z., Efficient Perovskite Light-Emitting Diodes by Buried Interface Modification with Triphenylphosphine Oxide, ACS Appl. Mater. Interfaces 2023, 15, 3644–3650. https://doi.org/10.1021/acsami.2c19123

Round 2

Reviewer 1 Report

Comments and Suggestions for Authors

Thank you for the author's response,I think this work can be accepted  in present form.

Reviewer 4 Report

Comments and Suggestions for Authors

The manuscript can be accepted in its current state.

Reviewer 5 Report

Comments and Suggestions for Authors

I acknowledge the author's response, and I'm satisfied with the author's efforts and appreciate the results harvested by the team. Considering the above, I suggest accepting this article in present form.

All the very best.

Comments on the Quality of English Language

Authors can proofread the entire manuscript and do minor grammar and format corrections.